

# Serum uric acid level is correlated with the clinical, pathological progression and prognosis of IgA nephropathy: an observational retrospective pilot-study

Pingfan Lu, Xiaoqing Li, Na Zhu, Yuanjun Deng, Yang Cai, Tianjing Zhang, Lele Liu, Xueping Lin, Yiyan Guo and Min Han

Division of Nephrology, Tongji Hospital, Tongji Medical College, Huazhong University of Science and Technology, Wuhan, China

## ABSTRACT

**Objectives**. This study was aimed to assess the relationship between serum uric acid (SUA) level and the clinical, pathological phenotype of IgA nephropathy (IgAN), and to determine the role of SUA level in the progression and prognosis of IgAN.

**Methods**. A total of 208 patients with IgAN were included in this study, and were classified into the normo-uricemia group and hyperuricemia group according to the SUA level. The clinical data at baseline, IgAN Oxford classification scores (MEST-C scoring system), and other pathological features were collected and further analyzed. All patients were followed up and the prognosis was assessed using Kaplan-Meier survival curves. GraphPad Prism 7.0 and SPSS 23.0 were used for statistical analyses.

**Results**. In clinical indicators, patients with hyperuricemia had the significantly higher proportion of males to females, mean arterial pressure, the levels of total cholesterol, triglyceride, Scr, BUN, 24 hour-urine protein, C3, and C4, the lower levels of high-density lipoprotein cholesterol and eGFR than those without ($p < 0.05$). In terms of pathological characteristics, the tubular atrophy/interstitial fibrosis scores, vascular injury scores, and glomerular sclerosis percentage were significantly higher in patients with hyperuricemia compared with those without ($p < 0.01$). There was no significant difference in the scores of mesangial hypercellularity, endocapillary hypercellularity, focal segmental glomerulosclerosis, as well as crescents between the two groups ($p > 0.05$). As for the depositions of immune complexes deposition in IgAN, the hyperuricemia group had less deposition of immunoglobulin G and FRA than the normo-uricemia group ($p < 0.05$), while the deposition of immunoglobulin A, immunoglobulin M, and complement C3 in the two groups showed no statistical difference. The survival curve suggested that patients in the hyperuricemia group have significantly poorer renal outcome than those in the normo-uricemia group ($p = 0.0147$). Results also revealed that the SUA level is a valuable predictor of renal outcome in patients with IgAN. The optimal cutoff value was 361.1 μmol/L (AUC $= 0.76 \pm 0.08167$) and 614 μmol/L (AUC $= 0.5728 \pm 0.2029$) for female and male, respectively.

**Conclusions**. The level of SUA is associated with renal function level and pathological severity of IgAN, and maybe a prognostic indicator of IgAN.

Corresponding author
Min Han, minhan@tjh.tjmu.edu.cn

# INTRODUCTION

IgA Nephropathy (IgAN), whose feature is the deposition of IgA-dominant in the mesangial area, is the most prevalent primary glomerular disease in the world. Approximately 40% of patients almost completely lose their renal function within 30 years after diagnosis (*Coppo & D'Amico, 2005*; *D'Amico, 2004*). Previous studies have demonstrated that clinical features, including severe proteinuria, poor renal function, and hypertension at the initial diagnosis are the predictors of poor outcomes in IgAN (*Le et al., 2012*; *Barbour & Reich, 2018*; *Mariani et al., 2018*; *Barbour & Reich, 2012*). Histological features were also identified as crucial predictors of IgAN patients. The Oxford classification of IgAN, also known as MEST score, including mesangial hypercellularity (M), endocapillary hypercellularity (E), segmental glomerulosclerosis (S), tubular atrophy/interstitial fibrosis (T) has been shown to be of significant value in predicting the prognosis IgAN (*Barbour et al., 2016*; *Maixnerova et al., 2016*; *Herzenberg et al., 2011*; *Katafuchi et al., 2011*; *Coppo et al., 2014*; *Lv et al., 2013a*). Recently, crescents (C) have been proposed to add to the Oxford classification of IgAN to form an updated MEST-C score system, which can provide a more comprehensive pathological prediction for the prognosis of IgAN (*Trimarchi et al., 2017*).

Numerous studies have shown that the level of uric acid can predict the incidence of atherosclerosis (*Feig, 2014*; *Gustafsson & Unwin, 2013*; *Li et al., 2014*), hypertension (*Wang et al., 2014*), and coronary heart disease (*Kim et al., 2010*; *Braga et al., 2016*). Moreover, some studies have emphasized that high levels of serum uric acid (SUA) would form urates crystals that deposit in renal tubules and interstitial, leading to kidney fibrosis and failure (*Su et al., 2020*; *Viggiano et al., 2018*). Recently, some studies have found that there is a correlation between the SUA level and the progression of IgAN. A cohort study has shown that hyperuricemia is associated with the progression of IgAN; however, the effect of hyperuricemia on renal pathology was not evaluated this study (*Bakan et al., 2015*). The study conducted by *Nagasawa et al. (2016)* has shown that the SUA level is a predictor of IgAN in females but not in males. Similarly, the evaluation of renal pathological changes was not included. *Moriyama et al. (2015)* have reported that hyperuricemia is a risk factor for the progression of IgAN with CKD stage G3a but not for stage G1, G2, or G3b-4. This study also has shown that, except for glomerulosclerosis percentage, there is no significant difference in the 2009 Oxford classification, crescentic percentage, and focal segmental sclerosis between the hyperuricemia group and the normo-uricemia group (*Moriyama et al., 2015*). Another study suggested that the relationship between hyperuricemia and IgAN progression was not very significant in patients with older age, lower eGFR, or interstitial lesion (*Zhu et al., 2018*). Similarly, this study did not use the updated Oxford classification of IgAN. These studies are partially inconsistent even contradictory and not comprehensive, cannot clarify the role of the SUA level in the progression and prognosis of IgAN. Therefore, it is imperative to conduct further research.

In this retrospective study, we first used the updated Oxford classification criteria (MEST-C), vascular injury, glomerulosclerosis, immunofluorescence score, clinical indicators, and prognosis analysis to comprehensively evaluate the relationship between the SUA level and the clinical, pathological characteristics of IgAN, to determine the impact of hyperuricemia on the development and prognosis of IgAN.

## METHODS

### Patients screening and clinical data collection

From January 2015 to May 2016, a total of 240 patients diagnosed with IgAN according to diagnostic criteria at Tongji Hospital were included in the study. The exclusion criteria of this study were listed as below: (1) the secondary IgAN caused by systemic diseases such as autoimmune disorders, chronic hepatitis, tumor, (2) incomplete clinical and pathologic data, or (3) the number of glomeruli in renal biopsy specimen less than eight. According to the above criteria, 32 patients were excluded and 208 patients were eventually involved in our research. The clinical data were collected at IgAN diagnosis. Blood samples were collected in the morning from fasting participants who had been directed to avoid eating any food that might affect the test result. The SUA levels >420 μmol/L in men and >360 μmol/L in women was defined as hyperuricemia. Based on the levels of SUA, there were 138 cases with hyperuricemia and 70 cases with normo-uricemia.

This study was approved by the Ethical Committee of Tongji Hospital (No.TJ-IRB20180608), and the Ethics Committee waived the need for informed consent from participants of this study.

### Histologic evaluation

Every renal biopsy specimen was scored by two pathologists who did not know the patient's clinical data. All of the renal specimens were classified and graded by five key pathological features: M, E, S, T, and C according to the MEST-C score of Oxford Classification (*Trimarchi et al., 2017*). Other pathological features, including glomerular global/ ischemic sclerosis, arteriosclerosis, arteriolar hyalinosis, and immune complex deposition were also described in all specimens. The histopathological grading schema we used was presented in Table 1. When the two pathologists differed in their assessments, the biopsy specimen must be reviewed again until an agreement was reached.

### Statistical analyses

GraphPad Prism 7.0 and SPSS 23.0 were used for statistical analyses. Continuous and categorical data were presented as mean ± SD and number (%), respectively. The comparison between the normo-uricemia group and the hyperuricemia group using the parametric $t$-test or Mann–Whitney. The overall renal survival rate of IgAN was presented using the Kaplan–Meier curve and the difference between the two groups was compared using the log-rank test. The $p < 0.05$ was regarded as statistical significance in all tests.

**Table 1    The histopathological grading schema in IgAN.**

| Histopathological features | Definition |
|---|---|
| **Mesangial hypercellularity** | |
| M0 | Mesangial score[a] $\leq 0.5$ |
| M1 | Mesangial score $> 0.5$ |
| **Endocapillary hypercellularity** | |
| E0 | absent |
| E1 | present |
| **Segmental glomerulosclerosis** | |
| S0 | absent |
| S1 | present |
| **Tubular atrophy/interstitial fibrosis** | |
| T0 | $\leq 25\%$ |
| T1 | 26%–50% |
| T2 | >50% |
| **Crescents** | |
| C0 | absent |
| C1 | present in $\geq 1$ glomerulus |
| C2 | in >25% of glomeruli |
| **Arteriosclerosis** | |
| **0** | no intimal thickening |
| 1 | intimal thickened and <thickness of media |
| 2 | intimal thickened and >thickness of media |
| **Arteriolar hyalinosis** | |
| 0 | absent |
| 1 | 1–25% of arterioles present |
| 2 | 26%–50% of arterioles present |
| 3 | >50% of arterioles present |

Notes.

[a]The glomerular mesangial area is the area between the glomerular capillary loops and is composed of mesangial cells and mesangial matrix. The mesangial hypercellularity score is calculated according to the mesangial cell count in per mesangial area: the scores of <4, 4–5, 6–7, $\geq 8$ mesangial cells are 0, 1, 2, 3, respectively. The mean score of all glomerulus is the mesangial hypercellularity score.

# RESULTS

## Demographic and clinical data

The demographic and clinical data of 208 participants in this study were shown in Table 2. The ages of all patients ranged from 15 to 63 years old, with an average of $33.8 \pm 10.7$ years old. Among them, 82 (39.42%) were males, and 126 (68.68%) were females. The mean arterial pressure of patients was $94.0 \pm 11.7$ mm Hg, among which 23 patients had hypertension previously, 16 (7.69%) had poor blood pressure control, and 22 (10.58%) were taking antihypertensive medication. In this study, 14.42% of the patients had previously received ACEI and/or ARB treatment, and 4.81% underwent tonsillectomy previously. For all patients, the mean levels of serum albumin, 24-h urine protein, SUA, serum creatinine (Scr), and eGFR were $39.48 \pm 5.77$ g/l, $1.35 \pm 1.58$ g/d, $348.00 \pm 109.10$ μmol/l, $89.89 \pm 41.49$ μmol/l and $82.23 \pm 30.22$ ml/min/1.73 m$^2$, respectively.

**Table 2  The demographic and clinical data of patients.**

| Parameters | Results |
|---|---|
| Age (years; mean ± SD) | 33.8 ± 10.7 |
| Gender (male/female) | 82/126 |
| Mean arterial pressure (mm Hg; mean ± SD) | 94.0 ± 11.7 |
| Treated with antihypertensive drugs, n (%) | 22 (10.58%) |
| Taking ACEI and/or ARB), n (%) | 30 (14.42%) |
| Underwent tonsillectomy, n (%) | 10 (4.81%) |
| Hematuria, n (%) | 207 (99.52%) |
| Albumin (g/l; mean ± SD) | 39.48 ± 5.77 |
| Uric acid (μmol/l; mean ± SD) | 348.00 ± 109.10 |
| Serum creatinine(μmol/l; mean ± SD) | 89.89 ± 41.49 |
| eGFR (ml/min per 1.73 m$^2$; mean ± SD) | 82.23 ± 30.22 |
| 24-h urine protein (g/d; mean ± SD) | 1.35 ± 1.58 |

**Notes.**
eGFR was calculated with the CKD-EPI equation. Data are expressed as mean ± SD or number (%).

## The comparison of clinical characteristics between the normo-uricemia group and the hyperuricemia group

The comparison of baseline characteristics between the normo-uricemia group and the hyperuricemia group was presented in Table 3. There were 70 patients (33.65%) with hyperuricemia, including 36 males (51.43%). These results were similar to the previous study conducted by *Shi et al. (2012)*. Results showed that the mean arterial pressure of patients with hyperuricemia was significantly higher than those without (98.2 ± 12.5 mmHg vs 91.8 ± 10.7 mmHg, $p = 0.0002$). In terms of coagulation indicators, although the FDP level was higher in the hyperuricemia group (3.86 ± 1.38 mg/l vs 3.23 ± 0.74 mg/l, $p = 0.0006$), the APTT level was similar in both groups ($p = 0.7833$). As for blood lipids, hyperuricemia patients had an elevated level of total cholesterol, triglyceride, and decreased level of HDL-C ($p = 0.026$, $p = 0.0005$, $p = 0.0056$, respectively). However, the LDL-C levels of the two groups were similar ($p > 0.05$). This study also revealed the blood urea nitrogen (6.49 ± 1.95 mmol/l vs 5.12 ± 1.52 mmol/l) , Scr (109.00 ± 36.54 Âţmol/l vs 80.40 ± 40.58 Âţmol/l) and 24-h urine protein (1.81 ± 1.67 g/d vs 1.05 ± 1.27 g/d) in hyperuricemia patients were higher, while the eGFR (74.50 ± 29.52 ml/min/1.73m2 vs 98.98 ± 25.12 ml/min/1.73m$^2$) was lower than those in normo-uricemia patients (all $p < 0.05$). When comparing immunological indexes of patients between the two groups, no significant difference was found in the levels of IgA, IgG, and IgM (all $p > 0.05$). However, the level of C3 and C4 were significantly decreased in the normo-uricemia group ($p = 0.0055$; $p = 0.003$).

## The comparisons of Oxford classification for IgAN between two groups

Histological features were evaluated using the MEST-C score of the 2016 updated Oxford Classification as *Trimarchi et al. (2017)* have reported. The comparisons in histological manifestations between the hyperuricemia group and normo-uricemia group were shown

**Table 3** The comparison of clinical characteristics between normo-uricemia group and hyperuricemia group.

| Parameters | Normo-uricemia | Hyperuricemia | P |
|---|---|---|---|
| Patients(n) | 138 | 70 | – |
| Gender (male/female) | 46/92 | 36/34 | **0.012** |
| Age (years) | 33.7 ± 10.5 | 34.2 ± 11.2 | 0.7311 |
| Body weight (kg) | 59.7 ± 9.3 | 64.8 ± 12.9 | **0.0057** |
| MAP (mm Hg) | 91.8 ± 10.7 | 98.2 ± 12.5 | **0.0002** |
| White blood cells (*$10°9$/l) | 7.04 ± 1.85 | 7.55 ± 2.44 | 0.1248 |
| Hemoglobin (g/l) | 129.60 ± 16.75 | 130.30 ± 23.52 | 0.8214 |
| Platelet (*$10^9$/l) | 237.50 ± 62.07 | 239.10 ± 70.38 | 0.8704 |
| APTT(s) | 37.43 ± 3.66 | 37.57 ± 3.23 | 0.7833 |
| FDP (mg/l) | 3.23 ± 0.74 | 3.86 ± 1.38 | **0.0006** |
| Total protein(g/l) | 68.84 ± 7.08 | 67.80 ± 8.40 | 0.3497 |
| Albumin(g/l) | 40.03 ± 4.94 | 38.40 ± 7.05 | 0.0868 |
| Total cholesterol(mmol/l) | 4.53 ± 1.17 | 5.01 ± 1.55 | **0.0260** |
| Triglyceride(mmol/l) | 1.63 ± 1.02 | 2.54 ± 1.89 | **0.0005** |
| HDL-cholesterol (mmol/l) | 1.34 ± 0.49 | 1.14 ± 0.35 | **0.0056** |
| LDL-cholesterol (mmol/l) | 2.73 ± 0.95 | 3.03 ± 1.41 | 0.1606 |
| Uric acid (μmol/l) | 290.00 ± 66.27 | 462.40 ± 83.46 | **<0.0001** |
| Blood urea nitrogen (mmol/l) | 5.12 ± 1.52 | 6.49 ± 1.95 | **<0.0001** |
| Serum creatinine (μmol/l) | 80.40 ± 40.58 | 109.00 ± 36.54 | **<0.0001** |
| eGFR (ml/min per 1.73mm$^2$) | 98.98 ± 25.12 | 74.50 ± 29.52 | **<0.0001** |
| 24-hour urine protein (g/d) | 1.05 ± 1.27 | 1.81 ± 1.67 | **0.0015** |
| IgA(g/l) | 3.22 ± 1.15 | 3.11 ± 1.00 | 0.5328 |
| IgG(g/l) | 2.91 ± 0.26 | 2.95 ± 0.38 | 0.1506 |
| IgM(g/l) | 1.56 ± 1.19 | 1.42 ± 0.67 | 0.3222 |
| C3(g/l) | 0.90 ± 0.22 | 1.00 ± 0.24 | **0.0055** |
| C4(g/l) | 0.20 ± 0.07 | 0.25 ± 0.12 | **0.0030** |

**Notes.**
MAP, mean arterial pressure; APTT, activated partial thromboplastin time; FDP, fibrin degradation products. eGFR was calculated with the CKD-EPI equation. Bold P-values indicate a statistical significance of $P < 0.05$.

in Fig. 1. The M, E, and S scores in the two groups were similar ($p > 0.05$). However, hyperuricemia patients had higher T scores than normo-uricemia patients ($1.00 ± 0.87$ vs $0.64 ± 0.76$, $p = 0.0023$). In the previous Oxford study, crescent formation was not listed as an independent predictor of renal outcomes. However, Hernan Trimarchi and his working group supported crescents as a predictor of renal outcome and suggested that crescent (C) should be added to the MEST score. Therefore, we compared the crescent scores in two groups; no significant differences were found ($p = 0.4650$). When comparing arterial lesions of patients in the two groups, hyperuricemia patients had higher scores of arteriosclerosis ($0.99 ± 0.79$ vs $0.48 ± 0.71$, $p < 0.0001$) and hyalinosis ($1.21 ± 0.87$ vs $0.63 ± 0.76$, $p < 0.0001$) than those without. The glomerular sclerosis percentage in hyperuricemia patients was higher than that in normo-uricemia patients ($0.22 ± 0.20$ vs $0.12 ± 0.15$, $p = 0.0007$).

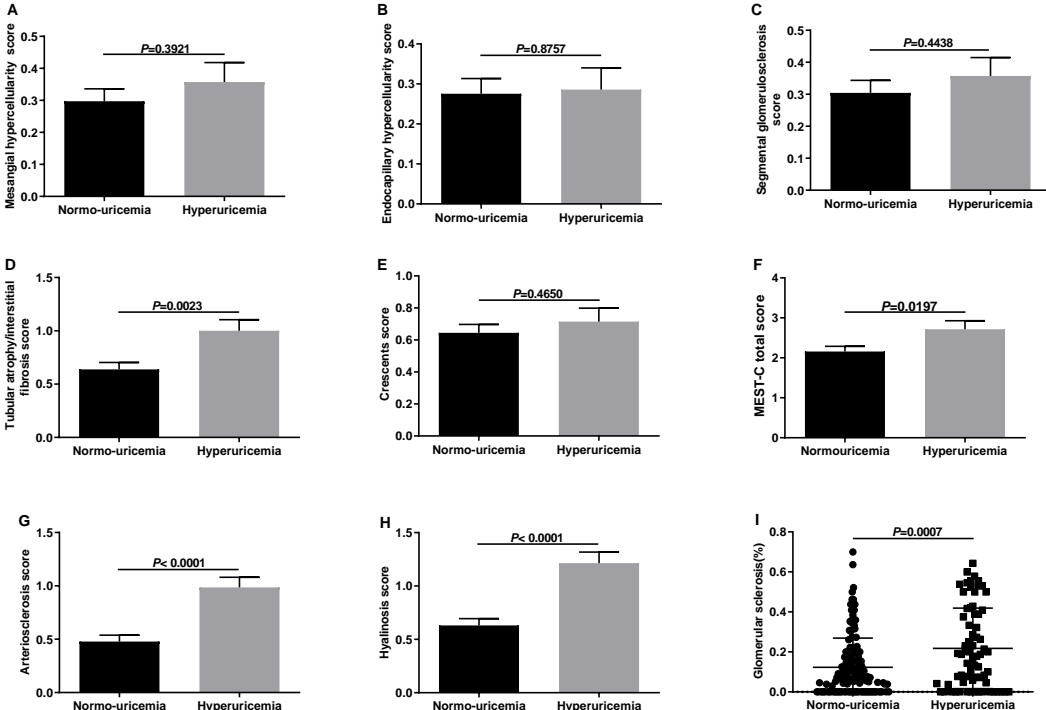

**Figure 1** **The comparisons of Oxford classification and arterial lesions between normo-uricemia and hyperuricemia patients in IgA nephropathy.** (A) Mesangial hypercellularity scores; (B) endocapillary hypercellularity scores; (C) segmental sclerosis scores; (D) tubular atrophy/interstitial fibrosis scores; (E) crescent scores; (F) MEST-C total scores; (G) arteriosclerosis score; (H) hyalinosis score; (I) the percent of glomerular sclerosis.

## The comparison of different types of crescent lesions between the normo-uricemia group and the hyperuricemia group

In the current investigation, results did not show any significant difference between the normo-uricemia group and hyperuricemia group in crescent (C) scores. Crescents could be divided into large and small crescents according to size and can be divided into cellular, fibrocellular and fibrous crescents according to crescent component (*Jennette, 2003*). Then we tried to explore whether SUA affected the formation of different types of crescents. As shown in Fig. 2, patients with hyperuricemia had slightly higher percentages of total crescents, large crescents, and small crescents than normo-uricemia, however, there was no statistical difference ($0.10 \pm 0.14$ vs $0.08 \pm 0.10$, $0.06 \pm 0.09$ vs $0.05 \pm 0.08$, $0.04 \pm 0.09$ vs $0.03 \pm 0.05$, respectively; all $p > 0.05$). Our data also showed that participants with hyperuricemia had a slightly higher percentage of fibrocellular ($0.05 \pm 0.09$ vs $0.04 \pm 0.07$, $p = 0.1916$) and fibrous crescent ($0.04 \pm 0.08$ vs $0.02 \pm 0.05$, $p = 0.0837$), while a significantly lower percentage of cellular crescent ($0.005 \pm 0.02$ vs $0.020 \pm 0.05$, $p = 0.0024$) compared to participants without hyperuricemia.

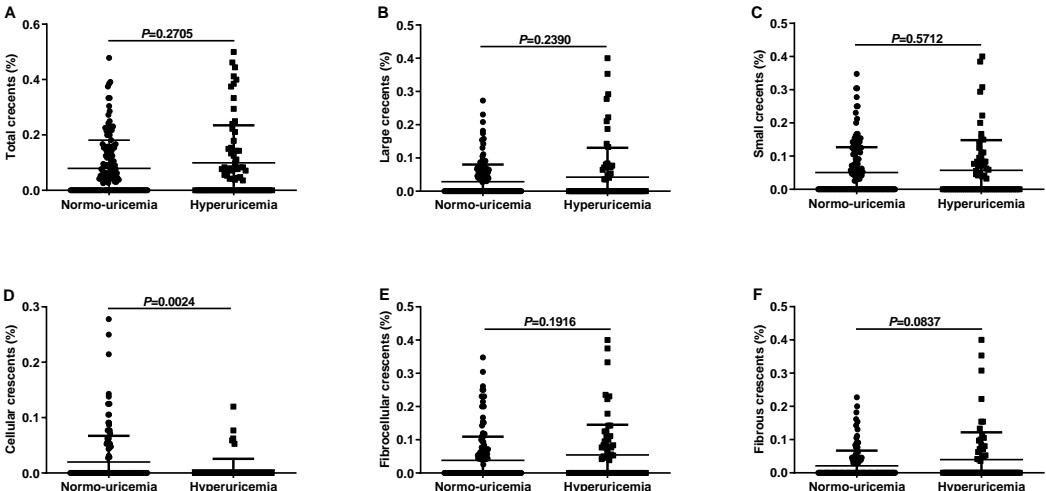

**Figure 2** **Percentages of different types of crescents among the IgAN patients with normo-uricemia or hyperuricemia.** The proportion of total crescents (A), large crescents (B), small crescents (C), cellular crescents (D), fibrocellular crescents (E), and fibrous crescents (F).

## The comparison of deposition of immune complexes and complements in renal biopsy between the two groups

The pathogenesis of IgAN was considered to be associated with immunological and or genetic mechanisms. Therefore, we evaluated the deposition of immune complexes and complements in the two groups. We classified the immune complex deposition according to the fluorescence intensity (FI) in this study. As shown in Fig. 3, the deposition of IgG and FRA in hyperuricemia group was less than those in normo-uricemia group (0.16 $\pm$ 0.44 vs 0.33 $\pm$ 0.63, $p = 0.0239$; 0.27 $\pm$ 0.51 vs 0.45 $\pm$ 0.55, $p = 0.0258$, respectively). No significant difference was detected in IgA, IgM, and C3 deposition between the two groups (all $p > 0.05$).

## The value of the SUA level in predicting renal prognosis

Finally, after an average of 25 months of follow-up, follow-up data were obtained from 193 patients. The endpoint of follow-up was defined as when patients needed dialysis treatment or when creatinine levels doubled. In this study, bout 70% of patients with HUA received uric acid-lowering drugs during the follow-up. The renal survival curves according to the uric acid level suggested that patients with hyperuricemia have poorer renal outcome than patients without ($p = 0.0147$, Fig. 4A). And then, we plotted a ROC curve to estimate the value of SUA in predicting the prognosis of IgAN. For females, the optimal cutoff value was 361.1 $\mu$mol/L, with sensitivity, specificity, and area under curve (AUC) of 0.7143, 0.7589, and 0.7679 $\pm$ 0.08167, respectively (Fig. 4B). However, for males, the optimal cutoff value was 614 $\mu$mol/L, with the sensitivity and specificity of 0.3333 and 0.9718. The AUC for male patients was 0.5728 $\pm$ 0.2029 (Fig. 4C). Our results revealed that the SUA level was a valuable indicator in predicting the prognosis of IgAN patients, especially in female patients, which is consistent with the research of *Oh et al. (2020)*.

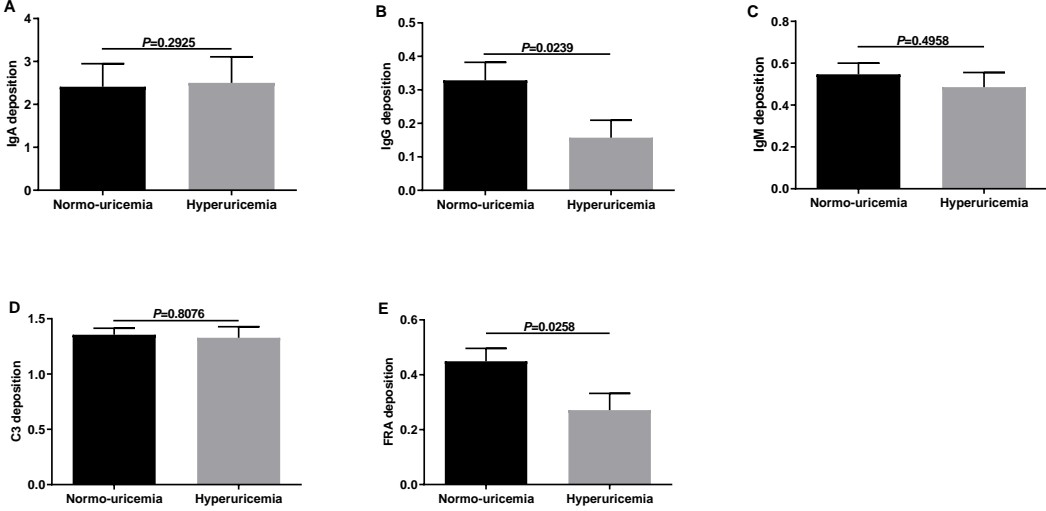

**Figure 3  The fluorescence intensity scores of immune complex deposits in renal biopsy.** Fluorescence intensity scores of IgA deposition (A), IgG deposition (B), IgM deposition (C), C3 deposition (D), and FR A deposition (E).

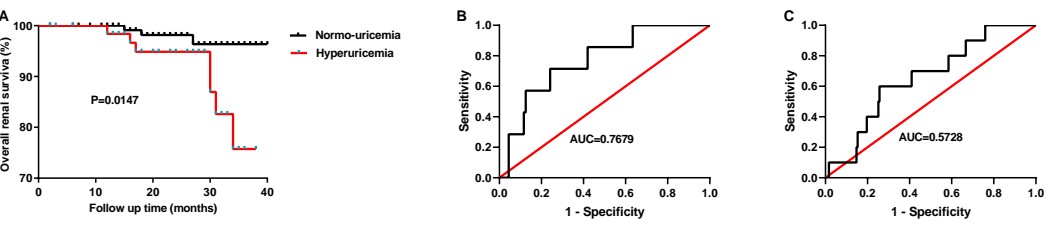

**Figure 4  The value of the serum uric acid level in predicting renal prognosis.** The overall renal survival rate of IgAN(A). ROC curve analysis evaluates the predictive value of uric acid level for the poor renal outcomes in females (B) and in males(C) with IgAN.

## DISCUSSION

IgAN has a high incidence and a variety of clinical manifestations, from asymptomatic hematuria to progress to renal failure rapidly with a few months. Consequently, it is imperative to ascertain its influencing indicator, delay its progression, and prevent ESRD. A few studies have demonstrated that the SUA level is closely related to the progression of IgAN (*Shi et al., 2012*; *Bakan et al., 2015*; *Cheng et al., 2013*). However, there are still inconsistencies and even contradictions in some studies. Therefore, predicting the prognosis of IgAN remains difficult due to its diverse clinical features. In this retrospective study of 208 IgAN patients, we comprehensively evaluated the correlation between SUA levels and clinical and histopathological features. Our results revealed that the SUA level can be considered as a predictor for the prognosis of IgAN.

Results showed that patients with hyperuricemia have a poorer renal function and higher blood pressure, which is consistent with Cui et al. researched on 148 patients with

IgAN (*Cui et al., 2011*). The results also revealed significant increases in blood pressure, lipid, C3, and C4 levels in patients with hyperuricemia. Previous studies have found that hyperuricemia is closely related to metabolic syndrome. Elevated SUA level has been confirmed to be an independent predictor of the development of diabetes, hypertension, hypertriglyceridemia, and nonalcoholic fatty liver disease (*Lv et al., 2013b*; *Grayson et al., 2011*; *Kuwabara et al., 2018*; *Yamada et al., 2010*). Some studies also showed a significant improvement in homeostasis assessment for the insulin resistance index after lowering the level of SUA (*Liu et al., 2015*; *Takir et al., 2015*). Previous researches revealed that the levels of serum C3 and C4 positively correlated with the development of the metabolic syndrome and had been identified as important markers relevant to this disease (*Meng et al., 2017*; *Meng et al., 2018*; *Xin et al., 2018*). The increased activation products of C3 can accelerate the uptake of free fatty acids, the synthesis of triacylglycerol, and inhibit hormone-sensitive lipase in several adipocytes, thus contribute to the development of the metabolic syndrome (*Phieler et al., 2013*; *Barbu et al., 2015*). However, the mechanism of C4 in metabolic syndrome remains unclear.

To evaluate pathological changes, we classified patients using the 2016 update Oxford classification of IgAN as described above. We found that among patients with hyperuricemia, 32.86% had M, 28.57% had S, and 35.71% had E, which was higher than those in normo-uricemia patients, data were 29.71%, 27.54%, 30.43% respectively. However, no significant difference was discovered in M, E, and S scores between the hyperuricemia group and normo-uricemia group. As for the tubular atrophy/interstitial fibrosis, this study revealed that the T score of hyperuricemia patients was statistically higher than that of normo-uricemia patients. This result demonstrated that the level of SUA was related to tubular atrophy/interstitial fibrosis, and hyperuricemia can be used as a clinical marker for T, which was consistent with the founding of *Myllymaki et al. (2005)* and *Fan et al. (2019)*. Recently, a study conducted by Nigro et al. found that the fractal dimension of tubules and the density of tubules negatively correlated the level of SUA and urea. They also suggested that SUA might be a better predictor to identify nephron integrity (*Nigro et al., 2018*). There were different views on the prognostic value of M, E, and S in different studies (*Coppo et al., 2014*; *Herzenberg et al., 2011*; *Shi et al., 2011*; *Kang et al., 2012*). Nevertheless, T was recognized as a prognostic indicator of IgAN in these studies. Thus, we regard the level of SUA as a valuable factor in predicting IgAN prognosis. The tubular atrophy/interstitial fibrosis caused by SUA might be explained by the following mechanisms. Urate crystals deposit in the renal tubules can directly damage or block the renal tubules, and also form uric acid renal stones to damage the kidney, which results in renal tubular atrophy and interstitial fibrosis, even renal failure (*Viggiano et al., 2018*). HUA could also promote the production of inflammatory factors such as MCP-1 and TNF-β1, which stimulates the inflammatory response and induce renal tubular injury and renal interstitial fibrosis (*Romi et al., 2017*). Moreover, HUA could decrease E-cadherin expression, increase $\alpha$-SMA expression, and induced tubular cells epithelial-mesenchymal transition, which results in renal tubulointerstitial injury (*Liu et al., 2017*).

Recently, the updated Oxford classification of IgAN recommends crescents (C) as a pathological predictor of renal outcomes in IgA nephropathy. In this current investigation,

no correlation between the SUA level and C score was detected. Then, we compared the percentages of different types of C lesions in patients with hyperuricemia and those with normo-uricemia and found that the percentages of total crescents, large crescents, small crescents, fibrocellular crescents, and fibrous crescents were slightly higher in hyperuricemia patients, while the percentages of cell crescents were significantly lower. It is well known that the formation of the crescent is caused by the deposition of immune complexes, the activation of monocytes, the release of inflammatory cytokines, and the aggregation of fibrocytes. The previous report discovered that uric acid might promote the release of inflammatory cytokines such as TNF-α, IL-1β, IL-6 in IgA patients (*Nakagawa et al., 2006*). Therefore, we speculate that hyperuricemia may be involved in crescent formation by promoting the release of inflammatory cytokines.

As shown in Fig. 1, hyperuricemia patients had significantly high percentages of glomerular sclerosis, including global glomerular sclerosis and ischemic glomerular sclerosis. Consequently, the level of SUA might indicate the severity of glomerular sclerosis. As reported in previous studies, many factors were related to glomerular sclerosis, including abnormal cytokine expression, podocyte injury or loss, and the activation of the RAS (*Riser, Cortes & Yee, 2000*; *Brown et al., 2000*). As mentioned before, SUA could promote the release of inflammatory cytokines, and persistent inflammation may result in glomerular sclerosis. Furthermore, hyperuricemia may activate the renin-angiotensin system, cause glomerular hypertension, and reduce perfusion, which ultimately led to the formation of glomerular sclerosis (*Sanchez-Lozada et al., 2005*). Besides, the uric acid can induce cellular oxidation through the xanthine oxidase pathway and then result in podocyte damage, which might be another cause of glomerular sclerosis.

We also evaluated the effect of elevated uric acid on vascular pathological changes, including arteriosclerosis and arteriolar hyalinosis. We found that the vascular lesions were more severe in hyperuricemia patients, and several mechanisms could explain this difference. Firstly, the elevated uric acid levels could inhibit nitric oxide synthase, induce inflammatory reactions, activate the renin-angiotensin system, cause endothelial cells dysfunction, and eventually lead to the lesions of vascular (*Behradmanesh et al., 2013*; *Corry et al., 2008*). Secondly, increased serum lipid levels and oxidation of LDL can promote the progression of atherosclerosis and arteriolar hyalinosis.

An early study by *Kim et al. (2012)*, which included 343 patients with IgAN found that the decreased serum C3 level and the deposition of C3 in mesangial could independently predict the progression of IgAN, indicating that the activation of complements was closely related to the IgAN pathogenesis. However, our result showed there was no significant difference in the deposition of IgA, IgM, and C3 between patients with hyperuricemia and those without. This could be due to the small sample size or different races of the study.

There are several limitations in this study. On the one hand, the sample size of this pilot study was relatively small. Next, we will conduct a multicenter, large sample size prospective study for further analysis. On the other hand, further studies are required to explore whether lowering uric acid levels could improve the outcome of IgAN.

In conclusion, the SUA level affects renal function and pathophysiology of IgAN. The levels of SUA can be considered as a prognostic indicator for IgAN, which is of great significance in guiding treatment decisions and assessing prognosis.

### Funding

This study was supported by the National Natural Science Foundation of China (No.81770686, 81970591) and the independent innovation research funds project of Huazhong University of Science and Technology (No.2017KFYXJJ101). The funders had no role in study design, data collection and analysis, decision to publish, or preparation of the manuscript.

### Grant Disclosures

The following grant information was disclosed by the authors:
National Natural Science Foundation of China: 81770686, 81970591.
Huazhong University of Science and Technology: 2017KFYXJJ101.

### Competing Interests

The authors declare there are no competing interests.

### Author Contributions

- Pingfan Lu conceived and designed the experiments, performed the experiments, authored or reviewed drafts of the paper, and approved the final draft.
- Xiaoqing Li conceived and designed the experiments, prepared figures and/or tables, and approved the final draft.
- Na Zhu and Tianjing Zhang performed the experiments, prepared figures and/or tables, and approved the final draft.
- Yuanjun Deng and Yang Cai performed the experiments, authored or reviewed drafts of the paper, and approved the final draft.
- Lele Liu analyzed the data, authored or reviewed drafts of the paper, and approved the final draft.
- Xueping Lin and Yiyan Guo analyzed the data, prepared figures and/or tables, and approved the final draft.
- Min Han conceived and designed the experiments, authored or reviewed drafts of the paper, and approved the final draft.

### Human Ethics

The following information was supplied relating to ethical approvals (i.e., approving body and any reference numbers):

This study was approved by the medical ethics committee of Tongji Hospital,Tongji Medical College, Huazhong University of Science and Technology (TJ-IRB20180608).

## Data Availability

The raw data including clinical data, histologic evaluation data, and follow-up data are available in the Supplemental Files.

## Supplemental Information

Supplemental information for this article can be found online at http://dx.doi.org/10.7717/peerj.10130#supplemental-information.

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
