# Peer review of "Serum uric acid level is correlated with the clinical, pathological progression and prognosis of IgA nephropathy: an observational retrospective pilot-study"

_PeerJ, doi:10.7717/peerj.10130_

## Round 0.1 · original submission · Major Revisions

Dear Dr. Han,

Your manuscript entitled “Serum uric acid level is correlated with the clinical, pathological progression and prognosis of IgA nephropathy: a retrospective study" which you submitted to PeerJ, has been reviewed by the editor and 3 external reviewers.

I regret to inform you that the reviewers have raised some significant concerns that need to be addressed before the manuscript can be considered further. However, since reviewers do find some merit in the paper, I would be willing to reconsider if you wish to undertake major revisions and resubmit.

The main criticism is the lack of novelty. According to PeerJ’s Editorial criteria, decisions are not made based on any subjective determination of impact, degree of advance, or novelty. However, the knowledge gap investigated by the research should be clearly stated, and statements should be made as to how the study adds value to the literature. Please also carefully consider the comments on the power of the study, statistics, and the use of SUA modifying medications.

Please also note that resubmitting your manuscript does not guarantee eventual acceptance. I must emphasize that the acceptability of the revision will depend upon the resolution of the points raised by the reviewers.

Sincerely yours,
Stefano Menini

Reviewer 1 ·

Basic reporting

Dr. Lu et al. studied the relationship between SUA and the clinical and pathological phenotype of IgA nephropathy. They evaluated 208 patients with IgAN divided in 2 groups, with normo-uricemia and hyper-uricemia and followed up them for an average of 25 months. Patients with hyperuricemia had higher mean arterial pressure, increased levels of total cholesterol and triglycerides and lower eGFR. When IgAN biopsies were evaluated using MEST-C score, the authors found out that patients with hyperuricemia had a higher scores of tubulointerstitial changes, arteriosclerosis, glomerular sclerosis and a significantly lower percentage of fibrocellular crescents. They concluded that the levels of SUA were associated with renal function level and pathological severity of IgAN and could have a prognostic relevance.
The language is clear and the literature well referenced. .

Experimental design

The figures and the tables are well labeled and described and the results are clearly explained. Methods are sufficiently described.

Validity of the findings

Several investigations have demonstrated the detrimental impact of SUA levels on the progression of IgAN, could the authors make clear the novelty of their findings?

Additional comments

I have some questions:
High blood pressure, increased levels of lipids and uric acid are some factors associated with metabolic syndrome. What about hyperglycemia and insulin resistance?

Have the patients received UA lowering agents, during the follow-up?

T score of patients with hyperuricemia was higher. This finding could be much more discussed.

Reviewer 2 ·

Basic reporting

The employment of the English language is good in general. although it could be improved.
References are adequately addressed, although the format is uneven.
The structure of the paper is adequate.
The results are tuned to the hypothesis.

Experimental design

The design is appropriate.
However, the power of the sampe has not been calculated for this sort of hypothesis to be proven. A 200-patient cohort is low.

Validity of the findings

The impact and novelty are low actually. I explain this topic to the authors.
Conclusions are well stated, but as the impact is low and lacks of novelty, in my opinion the paper offers nothing interesting.

Additional comments

This paper lacks of novelty. Thence, no high impact is expected.
As the authors hace demonstrated in a small cohort of 200 patients with biopsy-proven IgAN, high uric acid blood levels correate well with the chronic variables assessed. Hyperuricemia is a well-known cardiovascular risk factors for chronic situations are hypertension, left ventricular hypertrophy, hypercholesterolemia, CKD. We already know these findings.
With respect to IgAN, it is not unexpected to find that hyperuricemia is, again, correlated with chronic variables, as tubular atrophy and interstitial fibrosis, vascular damage, ang glomerulosclerosis.
Hyperuricemia is associated with CKD, and is a well-known predictor of renal outcome in CKD, glomerulopathies and IgAN.

Reviewer 3 ·

Basic reporting

The manuscript is well written. Please check for mistyping errors (seed e.g. table 1 legend "cells count in per")
References are in part outdated. Morphological measures in IgA nephropathy should be analyzed in terms of modern neprhon number techniques (see e.g. PMID: 31534861 ). The tubular changes also could be interpreted in view of techniques based on fractal analysis (PMID: 29540159). The role of urates could also be discussed taking care of more recent reviews (see e.g. PMID: 29689561).

Experimental design

aims are clear.
Statistics are not adequate: please correct for multiple testing.

Validity of the findings

Findings are interesting

Additional comments

Please clarify in Table 1 notes, what is a "mesangial area"
Table 2: uric acid - please check in parenthesis the units (umol/l or g/l?)
eGFR: please report in the table the equation used (CKD-Epi or MDRD etc)

Table 3: please verify the number of decimals reported in the table. For example, it is possibly not useful to report two decimals after the age (years) or to describe the weight (kg) or the MAP (mmHg) when the measure itself has a much lower precision

Figure 1 and 3: please use the same layout of Figure 2, reporting the values of each subject rather only the mean

---

## Round 0.2 · Minor Revisions

Dear Dr. Han,

Your manuscript entitled " Serum uric acid level is correlated with the clinical, pathological progression and prognosis of IgA nephropathy: a retrospective study " has again been carefully reviewed by the Editor and 3 Reviewers. Basically the revision is now acceptable for publication, but before final acceptance is given, I would appreciate it if you would address the remaining issue about the title of the manuscript raised by Reviewer 2.

If you are willing to do this, it would not be necessary for me to return the manuscript to the reviewers, but it could then be accepted for publication.

Sincerely yours,
Stefano Menini

Reviewer 1 ·

Basic reporting

The language is clear and the literature well referenced. The figures and the tables are well designed and described.

Experimental design

the results are clearly explained. Methods are sufficiently described.

Validity of the findings

the revised version strenghtened the findings

Additional comments

The authors mostly addressed my concerns.

Reviewer 2 ·

Basic reporting

The items to be addressed in this box are adequate.

Experimental design

Adequate

Validity of the findings

This topics hace been improved in the new version.

Additional comments

In general, the authors have considerabily improved the quality of the paper after the reviewers inputs.
In their letter to the editor and reviewers, after my comment that the number of patents in low, they comment that this study is "a pliot study", and that they plan to do a prospective trial.Thus, this statement must be addressed in the manuscript. Moreover, the title should be changed to: "Serum uric acid.....: an observational retrospective pilot-study".

Reviewer 3 ·

Basic reporting

In the revised manuscript the authors have responded to all my previous concerns

Experimental design

The experimental design is appropriate

Validity of the findings

The findings are interesting and will prompt further research about the subject

Additional comments

The revised manuscript presents novel and interesting material that is worth sharing with the scientific community

---

## Round 0.3 · accepted · Accept

Thank you for submitting a revised version of your manuscript. I am pleased to inform you that your manuscript is accepted for publication in PeerJ in its current form.

I thank all reviewers for their effort in improving the manuscript and the authors for their cooperation throughout the review process.

Sincerely yours,
Stefano Menini